# Non-invasive tape sampling of tryptophan and kynurenine in relation to phenylalanine and tyrosine from melanoma and adjacent non-lesional skin: A pilot study

Skaidre Jankovskaja[1,2]*, Peter Spégel[3], Kari Nielsen[4,5], Sebastian Björklund[1,2], Jeremy Bost[6], Johan Engblom[1,2], Gustav Christensen[4,5], Oksana Rogova[3], Maxim Morin[1,2], Merete Haedersdal[7,8], Martin Malmsten[8], Chris D. Anderson[9], Tautgirdas Ruzgas[1,2]*

**1** Department of Biomedical Science, Malmö University, Sweden, **2** Biofilms Research Centre for Biointerfaces, Malmö University, Sweden, **3** Centre for Analysis and Synthesis, Department of Chemistry, Lund University, Sweden, **4** Department of Dermatology, Skåne University Hospital, Lund, Sweden, **5** Department of Clinical Sciences Lund, Division of Dermatology and Venereology, Lund University, Lund, Sweden, **6** SciBase AB, Sundbyberg, Sweden, **7** Department of Dermatology, Copenhagen University Hospital – Bispebjerg, Copenhagen, Denmark, **8** Department of Clinical Medicine, University of Copenhagen, Copenhagen, Denmark, **9** Department of Biomedical and Clinical Sciences, Linköping University, Linköping, Sweden

* skaidre.jankovskaja@mau.se (SJ); tautgirdas.ruzgas@mau.se (TR)

## Abstract

### Purpose

To evade immunosurveillance many cancers convert tryptophan (Trp) into kynurenine (Kyn), which induces immunotolerance and suppresses immune responses. Elevated Kyn amounts have been found in blood from patients with cutaneous melanoma. This study aimed to investigate whether higher Kyn abundance and lower Trp abundance can be detected on the surface of cutaneous melanoma lesions compared with adjacent non-lesional skin.

### Methods

Sixteen patients with suspected melanomas were enrolled in this study. All lesions were excised and histopathologically diagnosed: 7 lesions were diagnosed as invasive malignant melanomas (MM), 6 as melanomas in situ (MIS), and 3 as benign lesions (BL). Non-invasive metabolite sampling was performed by tape stripping of suspected skin lesions and adjacent healthy non-lesional (NL) skin. Trp, Kyn, tyrosine (Tyr), and phenylalanine (Phe) were quantified by liquid chromatography-tandem mass spectrometry (LC-MS/MS). Electrical impedance spectroscopy (EIS) measurements were conducted to assess skin barrier integrity.

**Data availability statement:** All relevant data are within the manuscript and its Supporting Information files. S10 Table (excel file) contains study data set of measured variables and participant information.

**Funding:** The Knowledge Foundation (KK-stiftelsen), grant number 20190010; Stig and Ragna Gorthon foundation; EU Interreg ÖKS; Hudfonden (Welander Finsen foundation), the Regional research grant of the Southern health care region in Sweden; grant from the Swedish state under the agreement between the Swedish government and the county councils (the ALF agreement); the Inga and John Hain Foundation for medical research; the Eva and Ingemar Nilsson foundation; the Gyllenstiernska Krapperup Foundation; and the Gustaf Th Olsson foundation. The funders had no role in study design, data collection and analysis, decision to publish, or preparation of the manuscript.

**Competing interests:** The authors have declared no competing interests exist.

## Results

Levels of all metabolites, Tyr (x6), Phe (x6), Trp (x5), and Kyn (x3), were significantly higher in MM lesions compared with adjacent NL skin, resulting in an elevated Trp/Kyn ratio. Trp levels increased less than Phe and Tyr levels, suggesting a potential increase in Trp depletion. Skin resistance in MM lesions was half that of NL skin. No differences were observed between MIS or BL and NL skin.

## Conclusions

Non-invasive skin sampling revealed elevated Tyr, Phe, Trp and Kyn levels in MM skin, which is likely the result of compromised skin barrier at this stage of cutaneous melanoma.

## 1. Introduction

Skin cancer is among the five most common human malignancies with 2–3 million cases diagnosed each year worldwide [1,2]. Melanomas account for approximately 20% of skin cancers globally [3], and they represent the most aggressive and life-threatening form of skin cancer due to their ability to spread to other organs [4]. Early-stage melanoma detection is crucial for survival [4,5]. The standard melanoma diagnostic method is based on physical examination of suspected lesions followed by excisional biopsy and histopathological evaluation. The main disadvantage of this diagnostic approach is that it heavily relies on human judgment. Even the most experienced physicians may encounter difficulties in differentiating between early-stage melanomas and benign lesions, which makes diagnosis prone to false-positive, or even worse, false-negative results [6]. In addition, an excisional biopsy is an expensive, time-consuming procedure, involving surgery which bears its risks and discomforts [6]. Therefore, simple, non-invasive methods to assist with early melanoma detection, diagnosis, staging, prognosis, and monitoring of treatment response are in high demand.

An attractive strategy to aid melanoma diagnosis, easily executable on a high number of lesions, is non-invasive topical sampling of low molecular weight (LMW) metabolite biomarkers (molecular weight < 500 Da). These molecules can passively diffuse from viable skin layers to the skin surface within hours [7,8]. On the contrary, diffusion of high molecular weight (HMW) molecules, *e.g.*, proteins, across the skin is restricted. Proteins only reach the skin surface when keratinocytes differentiate into corneocytes, which takes 2–4 weeks [9]. Potential LMW biomarkers identified *in blood* from melanoma patients are mostly associated with melanogenesis, *e.g.*, 5-S-cysteinyldopa and, the L-3,4-dihydroxyphenylalanine (L-dopa)/L-tyrosine ratio [10]. However, several independent studies showed that melanoma patients have lower blood levels of Trp and higher levels of Kyn [11–13]. Kyn is produced from Trp in reactions catalyzed by indoleamine 2,3 dioxygenase 1 and 2 (IDO1, IDO2), and tryptophan 2,3-dioxygenase (TDO) [14,15]. Cells that express high levels of IDO deplete the microenvironment of Trp and replace it with its metabolite Kyn. Although the depletion of Trp is immunosuppressive, Kyn itself also has immune modulatory

properties [16]. For example, local depletion of Trp induces T cell starvation, causing reduced efficiency to clear out cancer cells, while the parallel buildup of Kyn facilitates cancer immune escape by reprogramming T-helper cells to T-regulatory cells [17–19]. Overall, lower concentrations of Trp and increased levels of Kyn in blood, *i.e.*, a lower Trp/Kyn ratio, indicate poor melanoma survival prognosis for a patient [11–13]. Although IDO1 is abundantly expressed in melanoma tissue [20–23], and IFN-γ-elicited over-expression of IDO1 reduces Trp and increases Kyn concentrations in *in vitro* skin models [24], the concentrations of Trp and Kyn in melanoma tissue remain to be assessed.

Several sampling methods could potentially be used to analyse Trp and Kyn in melanoma, including clinically used invasive techniques such as punch biopsy and fine needle aspiration (FNA), as well as non-invasive methods like tape stripping. However, to the best of our knowledge, no reports directly address Trp and Kyn concentrations in cutaneous melanoma tissue sampled by mentioned techniques or other methods. Punch biopsies provide full-thickness skin samples, which enable analysis of deeper tissue layers [25]. Though Trp and Kyn have not been measured in such samples, volatile metabolites [26] and lipids [27] have been successfully analysed in melanoma punch biopsies. Furthermore, studies analyzing Trp and Kyn in various models—including ex vivo diffusion studies across porcine skin [28], in vitro cell cultured skin models (measuring Trp and Kyn in both topical and basolateral compartments) [29], and investigations involving healthy human volunteers [30,31]—suggest that Kyn levels are higher in the deeper layers of the skin compared to the surface. This is important, as accurate measurement of Kyn at the skin surface is challenging. Therefore, sampling and analysis of metabolites in punch biopsies could potentially enhance reliability of Kyn quantification. Friendlier sampling FNA technique uses a thin needle to extract a small sample of tissue, which may contain cells, interstitial fluid, blood [32]. In the context of melanoma, FNA is commonly used to diagnose metastasis [33,34]. Metabolite analysis using FNA has been successfully demonstrated in thyroid cancer [35], suggesting its potential for sampling Trp and Kyn in melanoma tumors. However, FNA, as well as biopsies, is an invasive procedure and may present challenges in reproducibility due to tumor heterogeneity. In contrast, tape stripping is a non-invasive technique that avoids the risks associated with invasive sampling. It is particularly well-suited for repeated measurements and is effective for capturing surface-level metabolites, making it a promising tool for assessing Trp and Kyn levels directly on the skin surface. It is important to note that although tape stripping has not yet been applied to melanoma lesions to collect metabolites, it has been successfully used to collect genetic material for melanoma-related gene expression analysis, aiding in diagnosis [36–38]. Additionally, studies on psoriasis patients have shown that changes in metabolite profiles following treatment were detected earlier on the skin surface than in blood samples [39]. Concluding, punch biopsies and FNA might allow access to Trp and Kyn present in deeper skin layers, however tape sampling stands out as the most patient-friendly, non-invasive method and is particularly well-suited for suspected lesions screening and longitudinal studies.

The objective of this study was to investigate if a higher abundance of Kyn and a lower abundance of Trp can be detected on the surface of cutaneous melanoma compared with adjacent non-lesional skin. This investigation was conducted using tape sampling, offering a potentially valuable approach for metabolic profiling in melanoma.

## 2. Materials and methods

### 2.1 Materials

Tapes (D-squame®D100, 22 mm in diameter) and a standardized pressurizer (D500-D-squame®) from CuDerm (CuDerm, Dallas TX, USA) were used for skin surface sampling [31,40]. Trp, Kyn, Phe, Tyr, and isotopically labeled [$^2$H$_2$]- L-Phe were from Sigma-Aldrich (St. Louis, MO, USA). Formic acid (>99%) was purchased from Merck (Darmstadt, Germany). Isotopically labeled [$^{13}$C$_{11}$, $^{15}$N$_2$]-L-Trp, [$^2$H$_4$]-L-Tyr, and [$^{13}$C$_6$]-L-Kyn were from Alsachim (Illkirch-Graffenstaden, France). Methanol of LC-MS grade was purchased from VWR International (Fontenay-sous-Bois, France). Wet tissues soaked in 0.9% NaCl solution (Salvequick, wound cleanser) were from Orkla Care AB (Solna, Sweden). All water used was of Milli-Q grade (18.2 MΩ cm). Electrical impedance spectroscopy (EIS) measurements of the skin barrier were performed with Nevisense® (Nevisense 3.0, SciBase AB, Stockholm, Sweden).

## 2.2 Experimental design

This pilot study was conducted by the Consortium for Intelligent Diagnostics of Skin cancer (CINDIS) as a part of the BioMEL [41] research project approved by the Swedish Ethical Review Authority (Dnr 2013−101 with amendment Dnr 2022-02421-02), ClinicalTrials.gov ID NCT05446155. Sixteen patients with a lesion suspected to be melanoma and scheduled for surgery were recruited from the Department of Dermatology at Skåne University Hospital, Lund, Sweden between January and March of 2023. All included lesions were >6 mm in diameter. BioMEL inclusion and exclusion criteria were otherwise applied, as this was a pilot study with a limited number of participants. All participants gave written informed consent. Metabolite sampling, dermatoscopic images (Dermlite HÜD, Odense, Denmark) and EIS measurements were undertaken before excisional surgical standard procedures (according to national guidelines) for suspected melanoma lesions. All lesions were histopathologically assessed according to routine protocols. Metabolites were sampled from both the suspected melanoma lesion and adjacent non-lesional (NL) skin using tape-stripping, with a custom-made adhesive frame employed to define sampling area and minimize the inclusion of NL skin around the melanoma-suspected lesion. Lesional and NL skin were 2 cm apart (Fig 1; S1 Fig). Tape sampling was performed by pressing D-squame tapes onto the surface of the skin with a standardized D-squame pressurizer (225 g/cm$^2$ pressure, 22.24 mm area) for 5 sec and then quickly removing the tape using forceps. Three consecutive tapes were collected per sampling site. Each tape sample was collected into 1.5 mL Eppendorf tube and stored at −80 °C until analysis. Dermatoscopic skin images and EIS measurements were undertaken before and after the tape sampling.

## 2.3 Metabolite extraction

Metabolite extraction from the tape samples was performed according to the protocol from a previous study, in which matrix effects and analyte recoveries were evaluated [31]. Briefly, 1 mL of 20% methanol (v/v) in Milli-Q water was added to each tape sample and shaked for 1 h at ~400 rpm. After that, the tapes were removed and the extraction solution was centrifuged at 15 000 x g for 15 min. Supernatants were transferred to new Eppendorf tubes and stored at −20 °C. Before UPLC-MS/MS analysis, 200 µL of three consecutive tape extraction solutions were pooled, and 3 µL of a solution containing 5 µM each of isotope labelled Phe, Trp, Kyn, and 15 µM of isotopically labelled Tyr were added to achieve a final

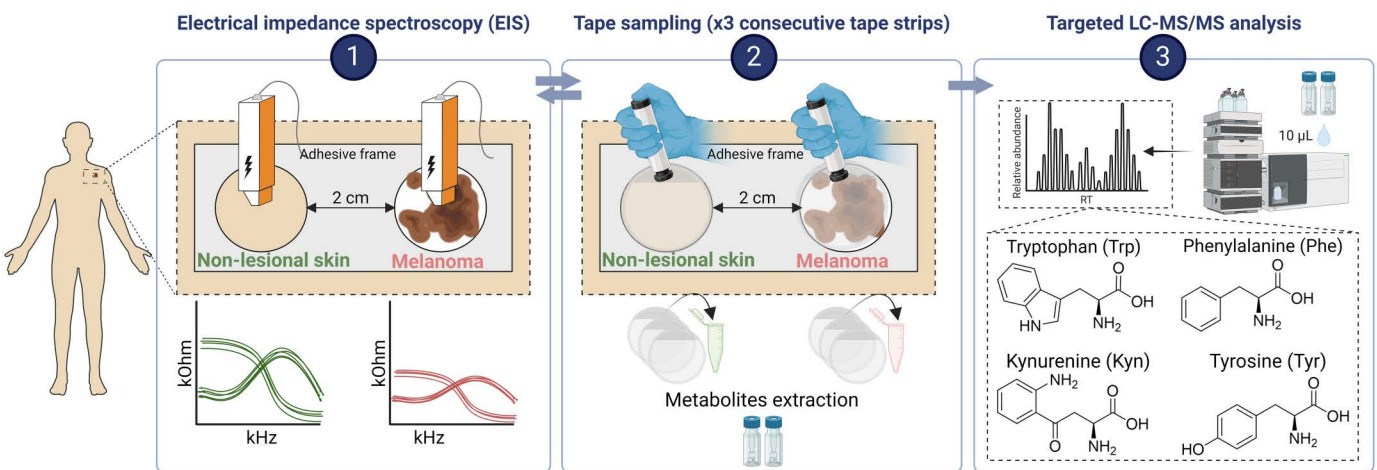

**Fig 1. Experimental design of the study performed on melanoma-suspected study participants.** A frame, designed to minimize the sampling of non-lesional skin along with melanoma-suspected lesion, was fixed on the skin, separating the two sampling areas by 2 cm. EIS measurements were conducted on both sampling areas prior tape sampling. Then, adhesive sampling with tapes was performed three times on each site with a new tape each time. After that, post sampling EIS measurements were conducted on the same lesions. LMW compounds were then extracted from the tapes and the abundance of four amino acids was determined by LC-MS/MS. Created with BioRender.com.

internal standard (IS) concentration of 0.25 µM for Phe, Trp, Kyn, and 0.75 µM for Tyr. Finally, 600 µL of tape sample was dried by miVac (SP Scientific, genevac, Ipswich, UK) and resuspended in 60 µL of 10% methanol in Milli-Q water. Important to note that different skin regions, NL skin and melanoma-suspected lesions, were assumed to provide similar and high analyte extraction recoveries. While this was not assessed in this study, we will refer to the following analysis as relative rather than absolutely quantitative.

## 2.4 LC-MS/MS analysis

Relative quantification of Tyr, Phe, Trp and Kyn was performed on an Agilent Infinity 1290 UPLC composed of a 1290 Infinity Binary Pump (G4220A, equipped with an Agilent Jet Weaver V35 mixer), a 1290 Infinity Multisampler (G7167B), and a 1290 Infinity Thermostatic Column Compartment (G1316C), coupled to an Agilent 6490 Triple Quadrupole mass spectrometer. 15 µL samples were injected and metabolites were separated using a reversed-phase C18 Kromasil column (250 x 4.6 mm, 5 µm, 100 Å from ES industries, New Jersey, USA). Metabolites were eluted using a 15 min gradient composed of solvent A (0.1% v/v formic acid in MilliQ water) and B (0.1% v/v formic acid in methanol) delivered at 0.8 mL/min: 10% B for 5 min, followed by a linear increase to 95% in 4 min, where it was held for 3 min, before being decreased to 10% in 0.1 min where it was held for 2.9 min. The column compartment was maintained at 22 °C. The mass spectrometer was operated in positive ESI mode, with the capillary voltage set at 4.0 kV, the nozzle voltage at 1.0 kV, the sheath gas temperature at 400 °C, the source gas temperature at 250 °C, the cone gas flow rate at 17 L/min, the sheath gas flow rate at 12 L/min, the nebulizer pressure at 40 psi, and the iFunnel high-pressure and low-pressure RF at 200 V and 100 V, respectively. Each analysis was carried out in dynamic multiple reaction monitoring (MRM) mode. The peak width for time filtering was set to 0.5 min, the cycle time to 500 ms, and the delta EMV (+) to 200 V. The MRM transitions and respective collision energies are shown in S1 Table.

Data analysis was performed in Skyline v3.5 software (MacCoss Lab Software, Seattle, WA, USA). Metabolite concentrations were determined using an external calibration curve in the range of 0.008–8 µM for Phe, Trp and Kyn, and 0.2–24 µM for Tyr spiked with the same IS concentrations as the tape extracts. Mixture of IS used for calibration curve preparation was same as the one used for the samples (see *2.3 Metabolite extraction* section). All samples, and calibration standards were prepared the same day, and measured in a single LC-MS/MS run. Each calibrant and sample were measured in triplicates. Repeatability of the analytical method, LOD and LOQ for each analyte is presented in S2 Table. Description of the preparation procedure of stock solutions and calibration standards can be found in S1 Section.

## 2.5 Skin barrier assessment by electrical impedance spectroscopy (EIS) measurements

EIS measurements were done to assess the skin barrier properties as described elsewhere [42,43]. Briefly, before EIS measurements, the skin surface was wetted with a saline tissue (0.9% NaCl) for 30 s, and excess saline was removed with a single-use cotton tissue. Four measurements, *i.e.,* pre- and post-sampling, were performed on each patient by using one single-use electrode probe. The measurements were performed by pressing the spring-loaded impedance probe (active area 0.5×0.5 cm²) against the skin site for approximately 8 s. In this work, we primarily focus on the absolute impedance at 1 kHz frequency (|Z| at 1 kHz), where skin impedance is dominated by the real part of the impedance and provides information about the resistive properties of the skin, which directly correlate with the integrity of the skin barrier residing in the outermost stratum corneum (SC) layer [44–46]. It should be noted that |Z| at 1 kHz in this work represents averaged values over all measured permutations/depths (S2 Fig). Other EIS parameters related to the electrical properties of the SC, such as the magnitude index (MIX) [44] and Max Phase were extracted from the EIS data (S2 Fig) and used for multivariate statistical analysis.

## 2.7 Statistical analysis

All statistical analyses were performed in R (version 4.3.3). Data is presented as box plots and reported as mean values ±SD. The boundaries of the boxplot represent the first and third quartiles, also known as the interquartile range (IQR). The thick bar inside the box represents the median, while the black square marks the mean value, and the whiskers show the range of data, excluding data beyond 1.5 times the IQR. These excluded values are displayed as individual dots and

are considered to be outliers. Statistical analysis was conducted on untreated data (raw data) and data without outliers for comparison; results presented in the main article are based on raw data unless stated otherwise. Plots were produced using the ''ggplot2''package. The normality of data was tested using the Shapiro−Wilk test (S3 Table). Normally distributed data of two related groups were compared using the paired Student´s t-test and non-normally distributed data by performing a non-parametric Wilcoxon signed-rank test. To correct for multiple testing false discovery rate correction (FDR) was applied. Comparisons across more than two groups were performed using one-way ANOVA, followed by a post-hoc Tukey test. For the count data, Fisher's exact test was performed. Correlations between analytes were evaluated by Spearman's rank test. Principal component analysis (PCA) was performed using "FactoMineR" package and visualized using "factoextra" package. Prior to the PCA analysis variables were mean centered and scaled to unit variance. Partial least squares-discriminant analysis (PLS-DA), extraction of scores, variable loadings, and variable importance in projection (VIP) was performed using ''ropls'' package and visualized using ''ggplot2'' and ''pheatmap'' packages.

## 3. Results

### 3.1 Study population and clinical diagnosis

Out of 16 study participants examined, 3 were ultimately defined to exhibit benign lesions (BL), diagnosed with dysplastic nevi or seborrheic keratosis. The remaining 13 patients had cutaneous melanoma, 6 of which were melanoma in situ (MIS) and 7 invasive melanomas, from now on termed malignant melanomas (MM). This classification of lesions is based on the pathologist's report on excised lesion samples, following the guidelines of the Union for International Cancer Control (UICC) 8th edition (2017) [47] and the American Joint Committee on Cancer (AJCC) 8th edition (2017) [48], which have been used in Sweden since January 2018 [49]. Characteristics of the melanoma patients included in the study are summarized in Table 1 and the lesion shown in Fig 2.

Table 1. Characteristics of the study participants. A more detailed description of diagnostic melanoma classification can be found in S2 Section.

| Patient no./sex | Age | Site | Sampling size[a] (mm) | Lesion size[b] (mm) | Diagnosis | Type | Stage | Breslow thickness (mm) | Infl |
|---|---|---|---|---|---|---|---|---|---|
| 02F | 38 | Back | 10 | 11x8 | MM | SSM | pT1a | 0.6 | ++ |
| 03F | 80 | Back | 12 | 15x10 | MM | SSM | pT1b | 0.9 | na |
| 06F | 75 | Thigh | 12 | 14x12 | MM | SSM | pT1a | 0.7 | na |
| 03M | 55 | Chest | 20 | 29x19 | MM | SSM | pT3a | 2.3 | na |
| 04M | 53 | Thigh | 12 | 18x15 | MM | SSM | pT3a | 3.7 | na |
| 08M | 61 | Back | 12 | 14x8 | MM | SSM | pT2a | 1.2 | + |
| 10M | 78 | Arm | 12 | 13x8 | MM | LMM | pT1a | 0.5 | na |
| 04F | 83 | Abdomen | 15 | 20x13 | MIS | – | – | – | + |
| 05F | 86 | Arm | 15 | 15x10 | MIS | – | – | – | ++ |
| 01M | 79 | Back | 8 | 6x8 | MIS | – | – | – | + |
| 05M | 48 | Arm | 7 | 7x5 | MIS[c] | – | – | – | ++ |
| 06M | 80 | Abdomen | 15 | 9x7 | MIS | – | – | – | + |
| 09M | 62 | Back | 7 | 7x5 | MIS | – | – | – | na |
| 01F | 42 | Back | 7 | 5x5 | BL[d] | – | – | – | na |
| 02M | 66 | Groin | 7 | 5x5 | BL[e] | – | – | – | na |
| 07M | 78 | Back | 10 | 11x7 | BL[d] | – | – | – | + |

Abbreviations: F – female; M – male; MM – malignant melanoma; MIS – melanoma in situ; BL – benign lesion; SSM – superficial spreading melanoma; LMM – lentigo maligna melanoma; pT - primary tumor: [a] – absence, [b] – presence of ulcerations; Infl – inflammation: [+]- present, [++] – high intensity, [-] absent, [na] – no information provided.

[a]Sampling size (diameter) defined by adhesive frame; [b]Lesion size extracted from biopsy report; [c]Dysplastic nevus, seborrheic keratosis and melanoma in situ; [d]Dysplastic nevus with severe dysplasia; [e]Seborrheic keratosis.

**Malignant melanoma (MM)**

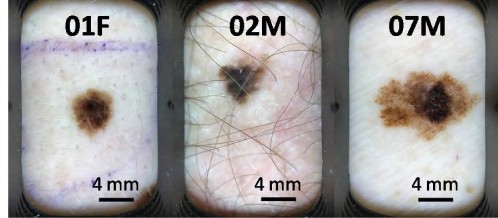

**Melanoma in situ (MIS)**

**Benign lesion (BL)**

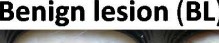

**Fig 2. Dermatoscopic images of melanoma-suspected skin lesions.** The study participants indicated on the top of the image correspond to patient names in Table 1.

The average age of study participants was 67 ± 16 years (mean ± SD, n = 16), and the sampling size, defined by the area of the sampling frame used, was 1.1 ± 0.7 cm² (n = 16). There were no statistically significant differences between females and males, and between differently diagnosed patient groups, i.e., BL, MIS and MM, in terms of age, sampling size, and site (S4 Table).

**3.2 Comparison between melanoma suspected skin lesion and non-lesional skin: the quantities of Trp, Kyn, Tyr and Phe, and their ratios**

Tyr, Phe, Trp and Kyn were well resolved using LC-MS/MS method (Fig 3).

Fig 4 presents the amounts of four metabolites and their ratios measured in the samples collected by tape sampling from the surface of melanoma-suspected lesions (BL, MIS, and MM) and adjacent NL skin. The methodological approach confirmed that three, consecutively taken tape samples (from a 1.1 cm² total sampling area, on average), pooled together, gave sufficient amounts of the amino acids to be quantified by LC-MS/MS. Collected amounts of Tyr, Phe, and Trp were in the range of 1–12 nmol/cm², while the collected amounts of Kyn were about three orders of magnitude lower, *i.e.*, 1–6 pmol/cm².

As can be seen in Fig 4A significantly higher amounts of Trp ($p = 0.006$), Tyr ($p = 0.009$), and Phe ($p = 0.006$) were collected from MM lesions compared with NL skin (S5 and S6 Table). For each patient, melanoma skin was compared with the adjacent normal (NL) skin, and the average change showed that approximately six times more Tyr and Phe and about five times more Trp were collected from MM skin compared with NL skin (S7 Table). Levels of Kyn tended to be about threefold higher in the MM lesions, with only one patient (08M) being an outlier, with higher Kyn levels on the NL skin.

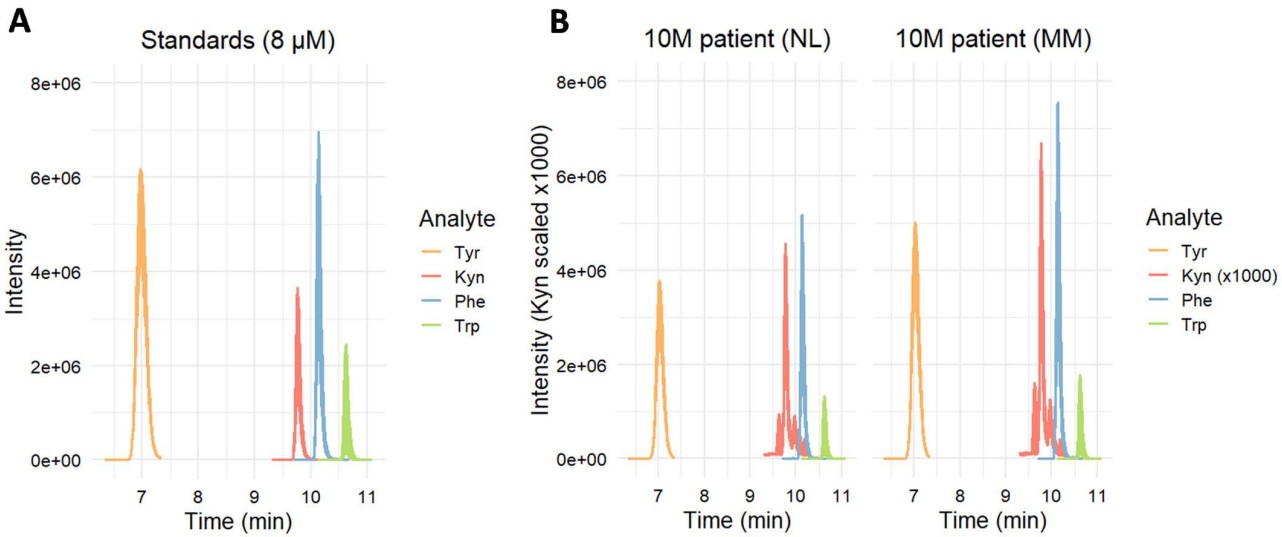

**Fig 3. LC-MS/MS chromatograms of a standard solution (A) and a patient sample (B).**

After removing Kyn values measured in the sample collected from the 08M patient, the difference between NL and MM became statistically significant (p = 0.015; S5 and S6 Table). In the case of MIS and BL patients (Fig 4A), no statistically significant difference was observed between the amounts of Tyr, Phe, Trp, and Kyn collected from MIS, BL lesions, and adjacent NL skin.

Comparing metabolite ratios, it can be stated that Trp/Tyr and Phe/Tyr ratios did not differ between lesional skin (MM, MIS, BL) and NL skin (Fig 4B, S5 and S6 Table). The Trp/Phe ratio determined in the samples collected from MM lesions was significantly lower compared with NL skin (p = 0.018). The Trp/Kyn ratio in MM lesions was significantly higher compared with NL skin (p = 0.028, S5 and S6 Table). Notably, the Trp/Kyn ratio, where Trp was normalized by Phe, decreases in the case of MM or MIS lesions compared with the adjacent NL skin (Fig 4B). Compared with the adjacent NL skin, the Trp/Phe and Trp/Kyn ratios were not significantly different in samples collected from MIS, or BL lesions.

To allow direct comparison of metabolite levels and their ratios across different skin surface samples, the data from NL skin, BL, MIS and MM lesions, was presented together in **Fig 5**.

Complementing the comparisons between lesional skin (BL, MIS, MM) and their adjacent NL skin shown in **Fig 4A**, **Fig 5A** clearly shows that collected amounts of Trp (p = 0.006), Tyr (p = 0.009), and Phe (p = 0.006) were significantly higher in MM lesions compared with all NL skin sampled in this study. However, similar to the comparison between lesional skin and adjacent NL skin, no significant difference was observed between MIS and all NL samples (**Fig 5A**). Interestingly, **Fig 5A** also illustrates that metabolites levels in BL are lower compared to MIS and MM lesions and are more similar to NL skin. Specifically, in MM lesions, the amount of Phe was significantly higher compared to BL (p = 0.004). While Tyr and Trp levels followed a similar trend, their differences were not statistically significant (Tyr: p = 0.089; Trp: p = 0.087). Summarizing, data shows that going from NL, MIS and MM lesion, there is an gradual metabolite levels increase. However, this trend does not hold for BL, which exhibits even lower metabolites levels than NL skin. In the case of Kyn, there was no statistically significant difference across NL, BL, MIS, and MM samples.

By comparing ratios determined across different skin samples (**Fig 5B**), statistically significant results were observed for the Trp/Kyn ratio, which was higher in MM lesions compared to all NL skin samples. Additionally, the Trp/Tyr and Trp/Phe ratios tended to be lower in melanoma lesions (MIS and MM) compared to benign skin (NL and BL). Specifically, the

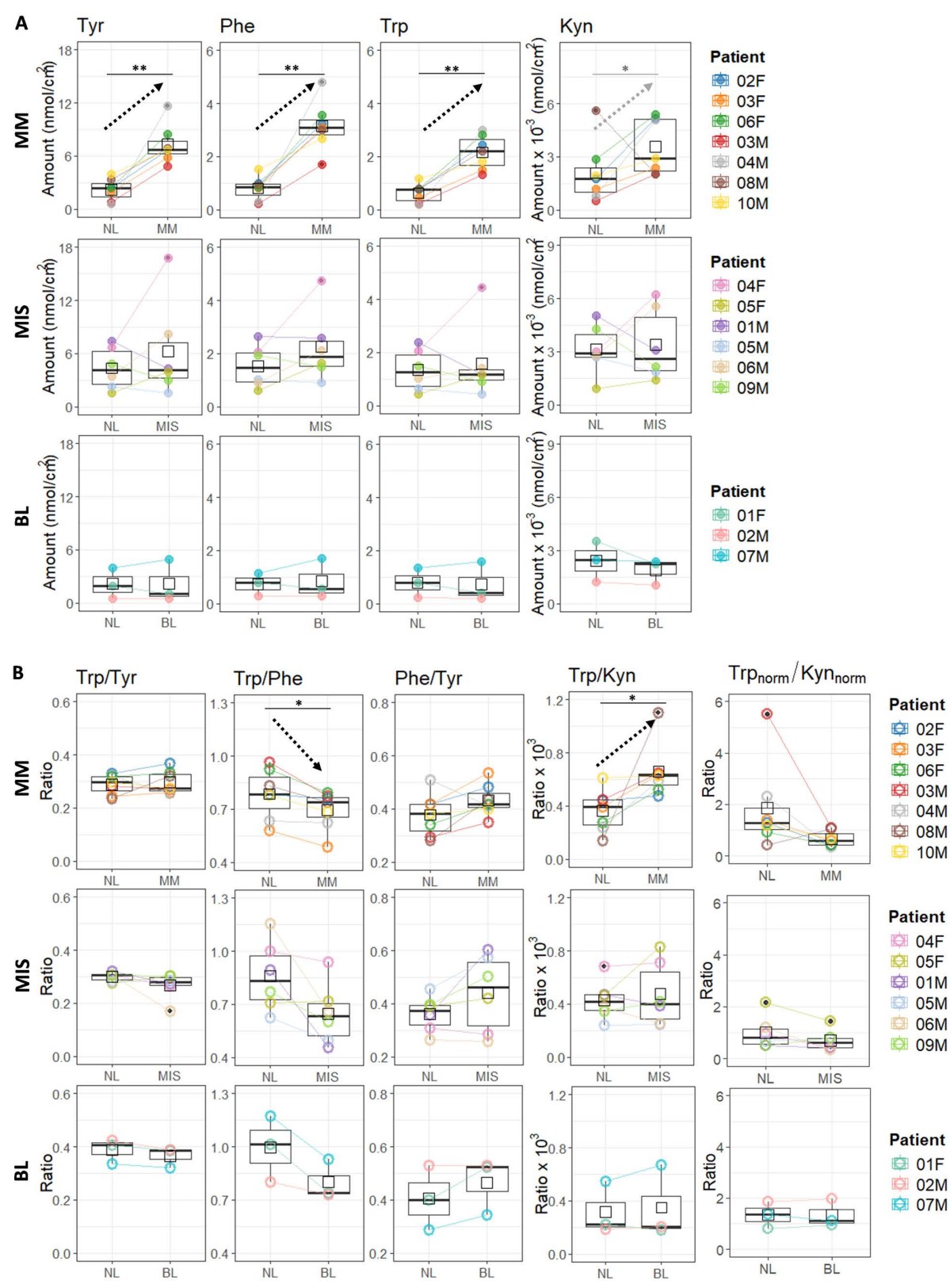

**Fig 4. The quantities of Tyr, Phe, Trp and Kyn (A), and their ratios (B), tape sampled from the surface of non-lesional (NL) skin compared with malignant melanoma (MM, n =7), melanoma in situ (MIS, n=6), and benign lesions (BL, n=3).** $Trp_{norm} = Trp/Phe$ and $Kyn_{norm} = Kyn/Kyn_{average}$. Comparison of melanoma-suspected skin lesions with non-lesional (NL) skin, was done by using paired two-sample t-test followed by FDR correction. Significance levels: '*' p<0.05, '**' p<0.01.

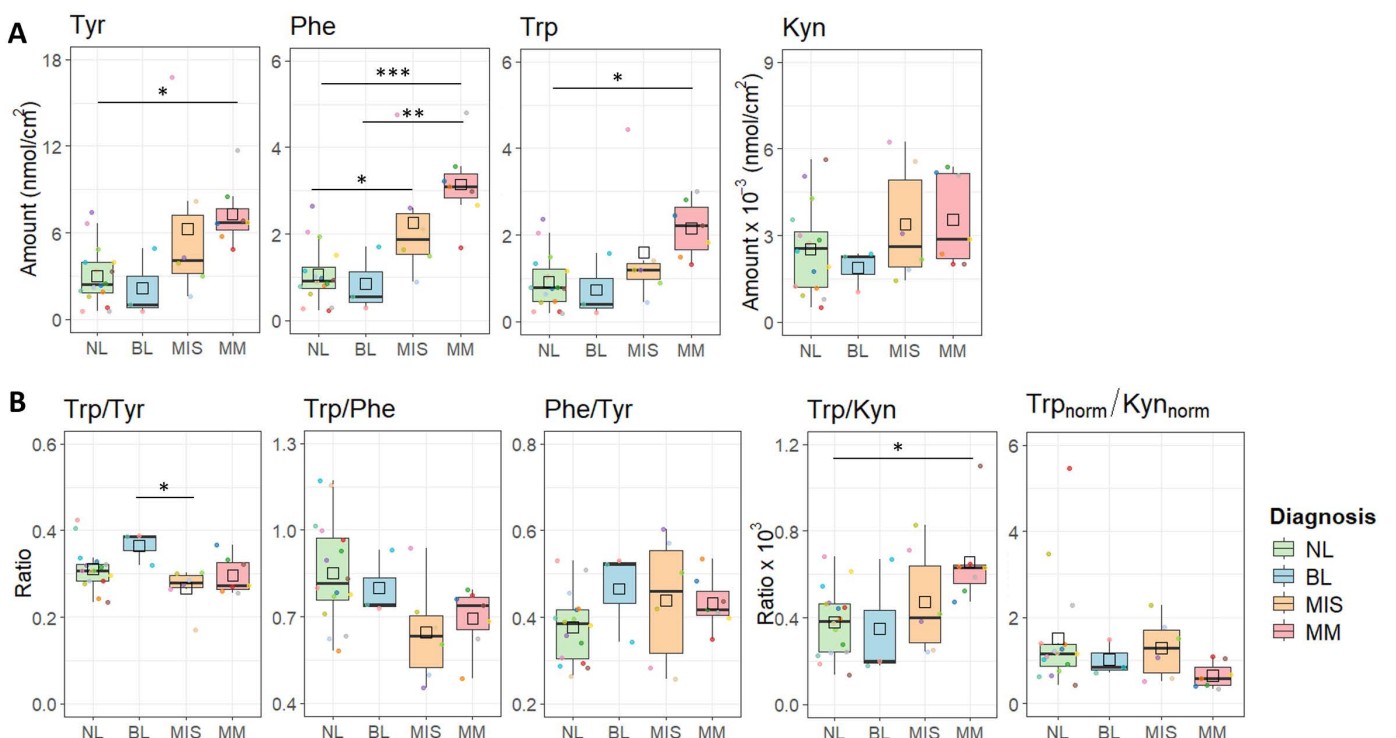

**Fig 5. Comparative analysis of metabolite quantities (A) and their ratios (B) across skin surface samples (NL, BL, MIS, MM).** Metabolite quantities and their ratios were compared across different skin surface samples using one-way ANOVA followed by a post-hoc Tukey test. Significance levels are indicated as: '*' p<0.05, '**' p<0.01, '***' p<0.001.

Trp/Tyr ratio was significantly lower in MIS lesions compared to BL (p=0.030), while the Trp/Phe ratio showed a trend toward being lower in MIS compared to NL (p=0.052).

### 3.3 Assessment of NL, BL, MIS, and MM skin barrier by resistance measurements

Skin barrier integrity was evaluated through EIS measurements, determining skin resistance, *i.e.*, impedance (IZI) at 1kHz, for both NL and lesional skin, before (pre-sampling) and after (post-sampling) tape sampling. The results are summarized in **Fig 6**.

EIS measurements conducted before tape sampling (pre-sampling) revealed significantly lower skin resistance values in MM lesions compared with adjacent NL skin, *i.e.*, 35.1±16.3 kOhm for MM and 80.3±30.4 kOhm for NL, n=7, p=0.017 (Fig 6A, S8 Table). Although BL and MIS lesions consistently showed lower skin resistance, both pre- and post-sampling, compared with the corresponding NL skin, the difference was not statistically significant, except for post-sampling BL (p=0.031) (Fig 6A, S8 Table). Removal of 3 consecutive tape strips from NL lesional skin decreased the skin resistance by 31% (p=0.006), whereas tape sampling on BL, MIS, and MM lesions did not significantly alter the skin resistance (Fig 6B).

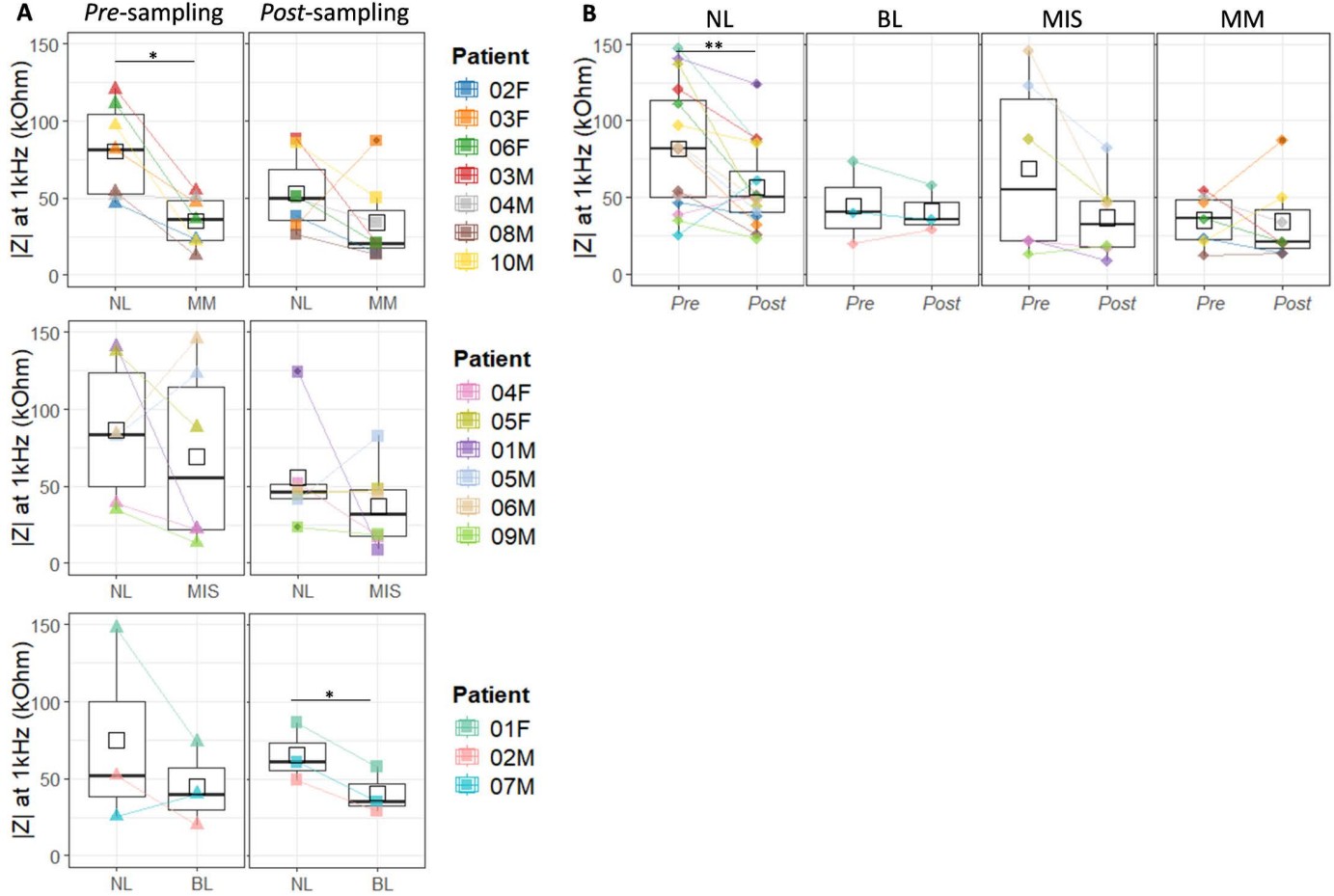

**Fig 6. Evaluation of skin barrier integrity using impedance at 1 kHz (skin resistance).** (A) Comparison of skin resistance between BL, MIS, MM lesions, and adjacent NL skin, measured before (pre-sampling) and after (post-sampling) tape sampling. (B) Effect of tape stripping on skin resistance, comparing measurements taken before (Pre) and after (Post) tape sampling. Comparison between two sample groups was done by using paired two-sample t-test test followed by FDR correction. Significance levels: '*' $p < 0.05$, '**' $p < 0.01$.

Correlations between skin resistance and analyte abundance showed a negative trend for pre-sampling NL skin, meaning that lower skin resistance was associated with higher amounts of analytes collected by tape sampling. Despite these observations, most of the correlations did not pass Spearman's significance test. The only significant result was for pre-sampling NL skin and sampled Phe levels, with $r = -0.53$ and $p = 0.047$ (S3 Fig).

### 3.4 Distinction between melanoma and non-melanoma skin

Along with the metabolites and skin barrier measurements evaluated and presented in Fig 4 and Fig 6, other parameters were available, *i.e.,* patient age, lesion size, and electrical properties of the skin barrier based on EIS measurements (S2 Fig). To explore the potential of all the measurements to differentiate between melanoma and non-melanoma skin samples, partial least square-discriminant analysis (PLS-DA) was utilized (Fig 7). Given the small size of our study cohort, a model discriminating between all four study participant groups, NL, BL, MIS and MM, was unsuccessful (S4 Fig). Instead, PLS-DA was employed to discriminate melanoma (M) samples consisting of MM and MIS patients, and non-melanoma (N) skin samples consisting of BL and NL skin (Fig 7). The latent variables 1 and 2 (LV1 and LV2) of PLS-DA analysis

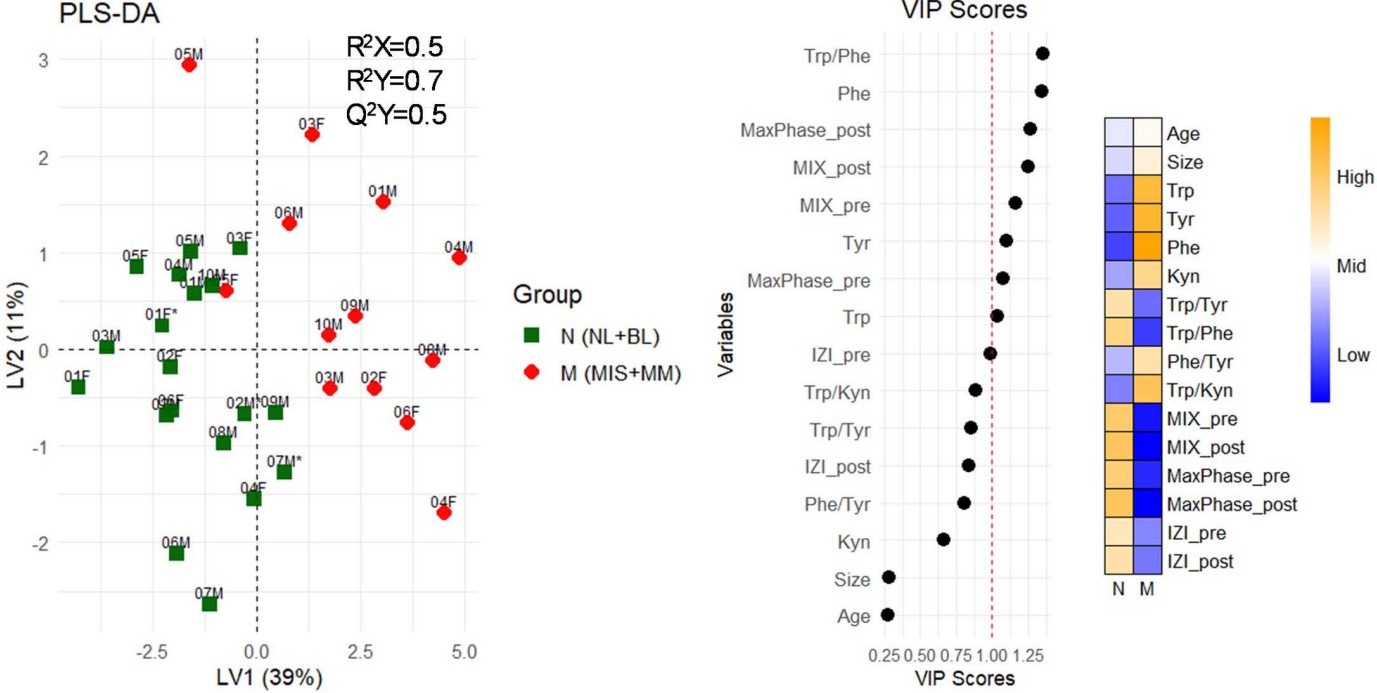

**Fig 7. Partial least squares-discriminant analysis (PLS-DA) for two patient groups: non-melanoma (N, including NL and BL samples) and melanoma (M, including MIS and MM samples).** Variables were ranked by variable importance projection (VIP) scores, where VIP scores > 1 reflect most important variables in separating between non-melanoma vs. melanoma sample groups. The heat map shows how variable levels (e.g., metabolite abundance, impedance values, lesion size) contribute to N and M groups separation.

accounted for 39% and 11% of the variance, respectively (Fig 7). The predictive ability of PLS-DA model was found to be moderate ($Q^2Y=0.5$), suggesting that the model has moderate ability to predict outcomes of unseen data (Fig 7). This PLS-DA model showed clear melanoma lesions separation from non-melanoma skin samples (Fig 7). Similarly, unsupervised multivariate principal component analysis (PCA) showed clustering of NL skin and MM lesions, whereas patients diagnosed with MIS and BL did not show a clear clustering pattern (S4 Fig). The VIP scores of PLS-DA indicates that the highest contribution for melanoma patient samples clustering is from the Trp/Phe ratio followed by the abundance of Phe, Tyr, and Trp, and pre- and post-sampling EIS measurements, *i.e.*, Max Phase, MIX and IZI at 1kHz (VIP ≤ 1). A heat map presenting the variable's contribution to N and M patient groups, provides a comprehensive overview of the results presented in Figs 4 and 6 and clearly illustrates that higher abundance of metabolites and lower skin barrier integrity is relevant for M patients, and opposite holds for N skin (Fig 7).

## 4. Discussion

### 4.1 Differential Trp and Kyn profiles relative to Phe and Tyr in benign, in situ and malignant melanoma lesions

Recent studies suggest that the conversion of the amino acid Trp to the immunosuppressive derivative Kyn within the tumor microenvironment (TME) contributes to the cancer's ability to evade the immune system [12,13,15,50]. Early and accurate melanoma detection remains crucial for improving patient outcomes [5], yet current diagnostic methods often rely on invasive biopsies, which can be limited by accessibility, cost, and patient discomfort [51]. Therefore, there is a growing need for non-invasive approaches to complement existing techniques. In this study, we have assessed the diagnostic

possibility of non-invasive sampling of Trp and Kyn to detect melanoma. The observations made in this study can be summarized for the case of MM and the case of MIS and BL.

**MM**. Quantification of Trp and Kyn in tape samples collected from MM lesions and adjacent NL skin showed that the amount of Kyn was three times higher in MM lesions. However, the significance of the difference in the amounts of Kyn collected from MM *versus* NL was only established after removing outlier data from patient 08M. Contrary to our expectations, Trp levels were significantly higher in MM lesions compared with NL skin. Moreover, even though Kyn levels increased in melanoma lesions, the Trp/Kyn ratio was higher owing to higher Trp abundance. Thus, the initial expectation that the Trp/Kyn ratio might be lower on the MM skin surface, as compared with NL skin, was not confirmed by our study. The possible explanation for this unexpected result could be the fact that Trp reaches the skin surface not only via its diffusion from viable skin layers, where cancer develops, but Trp is also a constituent of the natural moisturizing factor (NMF) pool generated from degradation of proteins, like filaggrin, in the SC [52,53]. Quantification of other NMF components such as Tyr and Phe, revealed that not only Trp amounts but also Tyr and Phe amounts were significantly higher in MM lesions than in NL skin. The levels of Phe and Tyr were six times higher, whereas Trp levels increased less, *i.e.*, five times. Normalizing the Trp/Kyn ratio by accounting for Phe abundance revealed a decrease in MM lesions compared to NL skin. This suggests that Trp depletion in the TME may be masked by Trp from the NMF pool, becoming apparent only when Trp levels are adjusted against another NMF component, such as Phe. MM lesions had more than two times lower skin resistance compared with adjacent NL skin, indicating a weakened and more permeable SC barrier. The permeability of the skin is a key parameter that can strongly influence the transport rate of LMW metabolites from the dermal compartments to the skin surface [7]. Considering that the skin permeability can be characterized by the electrical properties of the SC [54,55], higher amounts of metabolites collected from the skin surface may be due to more permeable skin. This finding is consistent with *in vitro* diffusion studies, where lower skin resistance correlates with higher skin permeability and thus faster diffusion of molecules from the viable epidermis towards the skin's surface [46,54–56]. Hence, increased amounts of analytes that can be sampled topically is expected in this case [7,54]. In contrast, a previous study on healthy human volunteers reported a weak but significant positive correlation between skin resistance and the amounts of Trp, Tyr, and Phe collected [30]. This could be because these amino acids contribute to the NMF pool, where a higher amounts of NMF may indicate a stronger skin barrier and, consequently, higher skin resistance. Therefore, it is important to distinguish between sampling from an intact and healthy SC tissue versus sampling from a compromised skin barrier. In addition, sampling from these two cases may differ in terms of tissue removal, where more corneocytes can be expected to be collected from skin lesions, leading to elevated metabolite extraction from MM samples compared with NL healthy skin.

Considering the factors of uncontrolled skin permeability, the amount of SC removed by tape sampling, and previous reports indicating that Tyr, Phe and Trp concentrations in TME [57,58] may be up to 10 times higher than in healthy tissue [58], the high abundance of amino acids on the surface of MM, may accurately reflect their concentrations in the TME of cutaneous MM tumors. To support this, a discovery metabolomics study conducted on low-metastatic melanoma (A375, G361), highly metastatic melanoma (A2058, SK-MEL-28) cell lines, and human epidermal melanocytes (HEM) showed that most amino acids, including Trp and Tyr, were present at higher levels in melanoma cell lines compared with epidermal melanocytes [59]. Similarly, using two melanoma cell lines derived from the metastatic and the primary sites of the same individual showed that the abundance of Trp and Tyr in the metastatic cell line was significantly higher compared with the primary melanoma cell line [60].

An additional factor that was not addressed in this study is the potential impact of skin microbiota on Trp and Kyn levels sampled from melanoma lesions. Alterations in the skin microbiota composition have been reported in melanoma [61]. Subsequently, skin microbiota may influence topical Trp and Kyn measurements in at least two ways. First, by altering the skin pH. The diffusion of permeant across SC depends on its hydrophilicity/charge, which is determined by pH driven protonation or deprotonation. Similarly, protonation/deprotonation of SC free fatty acids affects SC lipid packing and, thus, permeability through this barrier [62,63]. Second, microbiota can metabolize Trp and/or convert Trp to indoles, such as

indole-3-aldehyde, as shown in other studies [64,65]. Our earlier in vitro Trp and Kyn permeability measurements, conducted over 12 hours without antibacterial agents, demonstrated substantial microbial consumption of both metabolites—at rates exceeding passive diffusion [7]. Thus, incorporating skin microbiota analysis into future studies could greatly enhance the interpretation of topical metabolic profiles in melanoma lesions.

Concluding, although the exact origin of the highly elevated levels of Tyr, Phe, Trp, and Kyn on the surface of MM lesions is not completely clear, these levels may serve as easily accessible diagnostic biomarkers for MM.

**MIS and BL**. Regarding MIS and BL, neither the amounts of sampled amino acids nor skin resistance values were found to be statistically different. Amino acid ratios determined for MIS showed trends similar to MM, but this observation was not statistically significant. The small number of patients involved in this pilot study also did not allow us to arrive at a conclusive result regarding the diagnostic value of the Trp/Kyn ration or other amino acid ratios. Since MIS and BL occupy a diagnostically ''gray area" with a high risk of overdiagnosis and misdiagnosis [6,66], the lack of significant differences may be not only due to absent changes in early melanoma stages but also due to patient misclassification. For instance, BL patients 01F and 07M showed severe melanocyte dysplasia, which means these lesions can be also grouped with MIS lesions. Additionally, most MIS and BL lesions were smaller than the sampling frame (Fig 1; S5 Fig), introducing potential sampling errors that may have affected the results. To obtain more conclusive findings a study on a larger patient cohort should be performed, especially focusing on smaller size lesions, which are more relevant for clinical applications.

**Diagnostic value of multivariate statistics**. Both PCA and PLS-DA models showed clear clustering of melanoma samples, especially MM patients, with a relatively good separation from non-melanoma, *i.e.*, BL and NL skin samples. Similar sample clustering patterns observed in PCA and PLS-DA analysis support the validity of the MM and NL skin groups distinction and indicate that the group separation is robust. The discrimination between the two groups, non-melanoma and melanoma, strongly depends on the Trp/Phe ratio, on the abundance of Tyr, Phe, and Trp, and the skin barrier integrity as evaluated by EIS measurements. This demonstrates that both the physical properties of the skin barrier and the amino acids collected from the melanoma skin surface are valuable measures for cancer diagnostics. Overall, these general observations support further exploration of metabolic and skin barrier changes in melanoma development. With a larger patient cohort, noise-related variables could be more effectively identified and removed, potentially enabling the development of a model with higher predictive accuracy to improve melanoma patient diagnosis and outcomes.

### 4.2 Trp-Kyn pathway in skin and melanoma: contextualization with existing literature

To contextualize our findings, we reviewed prior studies examining Trp-Kyn metabolism in skin and melanoma using PubMed, with results summarized in S9 Table. Across more than 120 publications reviewed, a consistent theme emerges: under both pro-inflammatory and anti-inflammatory conditions, the Kyn pathway, particularly IDO1-mediated conversion of Trp to Kyn, is frequently upregulated. However, most of this evidence comes from in vitro experiments or animal models, with relatively few studies examining human melanoma tissues and no samples collected from melanoma skin surface (S9 Table).

A common view of Trp-Kyn metabolism in cancer immunomodulation is that it promotes immune suppression through two main mechanisms: metabolic competition, where Trp depletion leads to T cell starvation, and the accumulation of Kyn, which acts as a ligand for the aryl hydrocarbon receptor (AhR) [67]. The activity of Kyn pathway appears to be context-dependent. Depending on the inflammatory state or disease type, different enzymes, including IDO1, kynureninase (KYNU) metabolizing Kyn to anthranilic acid, kynurenine 3-mono-oxygenase (KMO) metabolising Kyn to 3-hydroxy-kynurenine etc., may be differentially expressed [68]. This suggests that focusing solely on Trp and Kyn may miss other downstream metabolites such as 3-hydroxykynurenine, 3-hydroxyanthranilic acid which, despite their low abundance,

could provide additional diagnostic insights [69]. Expanding the analysis to include these metabolites should be addressed in future studies.

Many studies report elevated Kyn and reduced Trp levels in blood of advanced melanoma patients [11–13], and at other inflammatory diseases [70]. A very few have measured these metabolites directly in tumor tissue of melanoma mouse model [15], with difficulty to find an equivalent study on human melanoma tissue (S9 Table). High number of studies done on cell culture models (S9 Table), while informative, cannot replicate the metabolic complexity of the in vivo microenvironment, where metabolite concentrations are shaped by interactions with different cells, vasculature, and immune infiltrates. Furthermore, while the Trp/Kyn ratio is often used as a surrogate for IDO1 activity, this relationship is not absolute, as IDO1 may also function through non-enzymatic signaling pathways [71,72]. The Trp and Kyn metabolism might be affected by other enzymes apart from IDO1 [73]. Direct quantification of these metabolites in tissue is thus essential for accurate interpretation, which currently is greatly lacking.

Therapeutically, the Trp-Kyn pathway has garnered interest as a target in melanoma, particularly in combination with immune checkpoint inhibitors such as programmed cell death protein 1 (PD-1) inhibitors. However, the key clinical trial on melanoma patients in Phase III using epacadostat (an IDO1 inhibitor) in combination with pembrolizumab (PD1 inhibitor) drugs, did not improve overall survival compared to placebo plus pembrolizumab [74]. Following these results, clinical trials using IDO1 inhibitors in other cancer types were also discontinued [75,76]. This setback has led to renewed efforts to understand cancer immune resistance mechanisms and to explore alternative strategies, such as enhancing Kyn degradation through enzymes like kynureninase [15,77] or blockage of AhR receptor [78], which have shown promising preclinical results in reducing tumor burden and prolonging survival.

While the prevailing view supports a central role for Trp-Kyn metabolism in immune evasion, some studies caution against overestimating its importance [79] relative to other metabolic pathways involved in tumor progression [73,80]. Our own findings contribute nuance to this picture. Although we observed elevated Kyn levels in melanoma lesions, Trp levels were even more increased if compared with adjacent healthy skin. Thus, Trp/Kyn ratio sampling from the surface of cutaneous melanoma lesions have not confirmed the decrease of the ratio on melanomas. Still, signs of increased Trp catabolism within the tumor microenvironment were apparent. Combining topical and tissue-level metabolite analysis may help clarify the spatial dynamics of Trp-Kyn metabolism in melanoma.

Finally, activation of Kyn pathway has been reported in other than melanoma inflammatory skin disorders, such as atopic dermatitis [81], allergic contact dermatitis [82], flaky paint (Pellagra-like) dermatosis [83], epidermolysis bullosa [84]. However, systematic Kyn levels in these conditions tend to show only mild increases, typically well below two-fold change. By comparison, our study observed an average three-fold increase in Kyn on the surface of MM lesions. Only a few studies report direct measurements of Trp-Kyn metabolites in/on defected skin (S9 Table). For example decreased Trp and elevated Kyn levels have been reported in hidradenitis suppurativa [85], while no differences were found between atopic dermatitis and healthy skin [65]. The lack of comparative data (S9 Table), and small sample size in our pilot melanoma study, limits our ability to differentiate melanoma-specific Trp-Kyn signatures from those related to non-malignant inflammation. Nevertheless, our non-invasive Trp-Kyn sampling and analysis represents a first step in assessing metabolite levels on the surface of cutaneous melanoma, though comprehensive studies are still needed to map and compare Trp-Kyn metabolism across skin layers and disease types.

## 5. Conclusions

This pilot study for the first time demonstrated the possibility of distinguishing MM lesions from NL skin by analysis of Tyr, Phe, Trp, and Kyn sampled from the skin surface. Non-invasive tape sampling revealed elevated Tyr (sixfold), Phe (sixfold), Trp (fivefold), and Kyn (threefold) levels in MM skin, which is likely the result of a compromised skin barrier. Because of the high levels of Trp found on lesional skin, the Trp/Kyn ratio was higher in MM lesions compared with non-lesional

skin. This finding contradicts the expectation that the Trp/Kyn ratio would decrease on the melanoma skin surface due to an upregulated Kyn pathway in the cancer TME.

## Supporting information

**S1 Fig.  Pictures of tape sampling.**
(DOCX)

**S1 Table.  Parameters for Tyr, Phe, Trp and Kyn quantification by LC-MS/MS in dynamic multiple reaction monitoring (MRM) mode.**
(DOCX)

**S2 Table.  Evaluation of analytical LC-MS/MS method performance.**
(DOCX)

**S1 Section.  Preparation of Tyr, Phe, Trp and Kyn stock solutions.**
(DOCX)

**S2 Fig.  Representative impedance data obtained by Nevisense measurement.**
(DOCX)

**S3 Table.  Normality assessment of paired differences using Shapiro-Wilk test.**
(DOCX)

**S2 Section.  Diagnostic melanoma classification.**
(DOCX)

**S4 Table.  Statistical evaluation of age, sampling size and site distribution between the study participants.**
(DOCX)

**S5 Table.  Average amounts of analytes and their determined ratios from study participants.**
(DOCX)

**S6 Table.  Compilation of p-values from statistical analyses.**
(DOCX)

**S7 Table.  Ratios between melanoma-suspected lesion and non-lesional skin.**
(DOCX)

**S8 Table.  Comparisons of skin resistance between the samples.**
(DOCX)

**S3 Fig.  Spearman's correlation between the amount of collected analytes and skin resistance.**
(DOCX)

**S4 Fig.  Principal component analysis (PCA) (a) and Partial least squares-discriminant analysis (PLS-DA) (b) of melanoma patients.**
(DOCX)

**S5 Fig.  Images showing sampling areas larger than melanoma lesions.**
(DOCX)

**S9 Table. Trp and Kyn in skin and melanoma: A PubMed literature overview.**
(XLSX)

**S10 Table. Study data set of measured variables and participant information.**
(XLSX)

## Acknowledgments

We thank research nurses Marie Sjögren and Carina Eriksson (Department of Dermatology, Skåne University Hospital, Lund) for their support with patient recruitment and sample collection; Stephan Bouman (Medicon Valley Alliance, Denmark) for CINDIS project management; and Annette Krais (Department of Chemistry, Lund University) for technical support with LC-MS/MS analysis.

## Author contributions

**Conceptualization:** Skaidre Jankovskaja, Peter Spégel, Kari Nielsen, Sebastian Björklund, Johan Engblom, Gustav Christensen, Merete Haedersdal, Martin Malmsten, Chris D Anderson, Tautgirdas Ruzgas.

**Data curation:** Skaidre Jankovskaja, Jeremy Bost.

**Funding acquisition:** Kari Nielsen, Johan Engblom, Tautgirdas Ruzgas.

**Investigation:** Skaidre Jankovskaja, Peter Spégel, Sebastian Björklund, Jeremy Bost, Johan Engblom, Gustav Christensen, Merete Haedersdal, Martin Malmsten, Chris D Anderson, Tautgirdas Ruzgas.

**Methodology:** Skaidre Jankovskaja, Peter Spégel, Oksana Rogova, Maxim Morin.

**Resources:** Kari Nielsen, Gustav Christensen, Merete Haedersdal, Tautgirdas Ruzgas.

**Supervision:** Peter Spégel, Kari Nielsen, Sebastian Björklund, Tautgirdas Ruzgas.

**Visualization:** Skaidre Jankovskaja.

**Writing – original draft:** Skaidre Jankovskaja.

**Writing – review & editing:** Skaidre Jankovskaja, Peter Spégel, Kari Nielsen, Sebastian Björklund, Jeremy Bost, Johan Engblom, Gustav Christensen, Oksana Rogova, Maxim Morin, Merete Haedersdal, Martin Malmsten, Chris D Anderson, Tautgirdas Ruzgas.

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
