## [Decision Letter · Decision Letter 0]

Dear Dr. Jankovskaja,

We look forward to receiving your revised manuscript.

Kind regards,

Krishnendu Sinha, Ph.D.

Academic Editor

PLOS ONE

Journal Requirements:

“The Knowledge Foundation (KK-stiftelsen), grant number 20190010; Stig and Ragna Gorthon foundation; EU Interreg ÖKS; Hudfonden (Welander Finsen foundation), the Regional research grant of the Southern health care region in Sweden;  grant from the Swedish state under the agreement between the Swedish government and the county councils (the ALF agreement); the Inga and John Hain Foundation for medical research; the Eva and Ingemar Nilsson foundation; the Gyllenstiernska Krapperup Foundation; and the Gustaf Th Olsson foundation.”

4. We note that Figure 1 in your submission contain copyrighted images. All PLOS content is published under the Creative Commons Attribution License (CC BY 4.0), which means that the manuscript, images, and Supporting Information files will be freely available online, and any third party is permitted to access, download, copy, distribute, and use these materials in any way, even commercially, with proper attribution. For more information, see our copyright guidelines: http://journals.plos.org/plosone/s/licenses-and-copyright.

Reviewers' comments:

Reviewer's Responses to Questions

**Comments to the Author**

1. Is the manuscript technically sound, and do the data support the conclusions?

Reviewer #1: Partly

Reviewer #2: Yes

2. Has the statistical analysis been performed appropriately and rigorously?

Reviewer #1: I Don't Know

Reviewer #2: Yes

3. Have the authors made all data underlying the findings in their manuscript fully available?

Reviewer #1: No

Reviewer #2: Yes

4. Is the manuscript presented in an intelligible fashion and written in standard English?

Reviewer #1: Yes

Reviewer #2: Yes

Reviewer #1: The paper describes an interesting approach to answering a very common question: Is this skin "defect" a melanoma or not ? And it would be very relevant if the described tape analysis could make it easier for patients to have a correct diagnosis. Furthermore - the tape test could potentially be offered as a test you perform at home and just send the tape samples to a laboratory for analysis.

However I have quite a few comments, particularly to the standardization of the analytical method.

1. Introduction

In the introduction (Line 37-38) it is stated that over a million cases of skin cancer is diagnosed each year worldwide. The reference for this statement is ref 1, which is from 2013. But ref. 1 refers back to a paper from 2010 that summarizes numbers from 2006. Is is possible to find more recent information about the number diagnosed per year ?

Ref. 3 - please check - it must be the wrong paper in the ref list.

Ref 4 - is that correct ? Please check all references carefully

2.3 Metabolite extraction

In section 2.3 it is described that the three tape samples are extracted for one hour with 1 mL 20% methanol. Then 200 µL from each extraction is pooled before addition of the internal standard. I know it is too late to change this but I would definitely have added the internal standards to the 1 mL extraction solvent to also compensate for losses during extraction.

2.4 LC-MS/MS/ analysis

In general I find it very difficult to deduct how the quantification was performed. Generally in LC-MS/MS analyses calibrators, internal controls and samples are prepared in the same way, i.e. you pipette a certain amount of sample and add a certain amount of internal standard to everything. Then the calibration curve is obtained (automatically in the LC-MS/MS system) by plotting the concentration on the X-axis and the response factor (peak area of analyte divided by peak area of internal standard).

In this paper internal standard (unknown amount of 0,25 µM) is added to combined extracts (se comment above too) of tape samples whereas the calibrators are prepared with the internal standard already added. The paper does not mention if calibrators are prepared and aliquoted to be used for different sample sets or if the calibrators are prepared every time samples are run. Or maybe the whole experiment was one single run ? Anyway if samples and calibrators were not prepared on the same day with the same solution of internal standard added in the same amount it could result in erroneous results.

Furthermore - as I read the paper - no internal controls are used. I appreciate that internal controls cannot be prepared in the same way as the samples since you cannot have multiple skin collection sites with known amount of the amino acids. But it would be interesting to see if you could extract a known amount of amino acids that were pipetted directly on the tape (in a small amount of liquid) and then extracted in the same way as the patient samples.

In Table S2 - please adjust the number of digits after the comma to more reasonable numbers.

The precision of the method seems to be a measure of repeatability rather than intermediary precision since it is a measurement of the same calibration solutions three times, as I read it. The precisions obtained are therefore very low and not comparable with intermediary precisions found in the normal way where you have internal controls that are measured together with the calibrators and samples over at least 5 different days from five different sample preparation sets.

The limit of quantification would normally be determined as a precision profile showing the lowest concentration that can be measured on internal controls near the LOQ with an intermediary precision below 20% in at least 5 sample sets. Here you do not have any internal controls - hence the calculation from the standard curve instead which may be acceptable. As regards the table showing the LOD and the LOQ - please explain the two numbers before and after the slashes ?

Having said all this the exact concentration may not be very important since extract from the lesions are compared with extracts from the same patient in a healthy area. But you need to describe in more detail, so that we know you are not seeing a systematic difference between lesion type just because the samples were from individual runs with potentially different concentrations of the isotopically labelled internal standards (due to lack of stability). I hope my point makes sense …

Results

In Table 1 there is a column called Sampling size which shows the diameter defined by adhesive frame. I am not sure how to understand this since earlier in the paper it is stated that the tape is 20 mm in diameter. So why is the sampling size less than 20 mm ?

Figures

Since the LC-MS/MS analysis has not been published previously I think you should add a Figure showing a chromatogram for for instance a calibrator and a patient sample with low concentrations of the analytes.

Reviewer #2: The study is designed and executed in methodologically and medically sound manner. Clustering of malignant and benign tissues in multivariate analysis is impressive. Discussion on biological reasons of different aminoacid signatures is also sound. In conclusion I would recommend this pilot study for publication.

**Do you want your identity to be public for this peer review?** For information about this choice, including consent withdrawal, please see our Privacy Policy

Reviewer #1: No

Reviewer #2: **Yes: ** Benjamin Benzon

---

## [Author Response · Author response to Decision Letter 1]

18 Feb 2025

[Thank you, editor and reviewers for the comments on our work. The manuscript has been corrected/improved accordingly. Reponse to Reviewers comments can be found as sperately uploaded word document within the submission.]

Reviewer #1: The paper describes an interesting approach to answering a very common question: Is this skin "defect" a melanoma or not ? And it would be very relevant if the described tape analysis could make it easier for patients to have a correct diagnosis. Furthermore - the tape test could potentially be offered as a test you perform at home and just send the tape samples to a laboratory for analysis.

However I have quite a few comments, particularly to the standardization of the analytical method.

1. Introduction

In the introduction (Line 37-38) it is stated that over a million cases of skin cancer is diagnosed each year worldwide. The reference for this statement is ref 1, which is from 2013. But ref. 1 refers back to a paper from 2010 that summarizes numbers from 2006. Is is possible to find more recent information about the number diagnosed per year?

ANSWER. Thank you for this comment. The information about diagnosed skin cancer cases per year was updated by using a recently published paper (Ferlay et al., 2024) and statistics from World Health Organization (WHO) website published in 2017. Please, see lines 37-38.

Reviewer #1. Ref. 3 - please check - it must be the wrong paper in the ref list. Ref 4 - is that correct ? Please check all references carefully

ANSWER. Thank you. It was updated to references focusing on primarily on melanoma aggressiveness, metastasis and early detection issue. Please, see newly added references number 4 and 5. Considering reviewer comment about the refences, all references have been checked, and updated.

Reviewer #1. 2.3 Metabolite extraction

In section 2.3 it is described that the three tape samples are extracted for one hour with 1 mL 20% methanol. Then 200 µL from each extraction is pooled before addition of the internal standard. I know it is too late to change this but I would definitely have added the internal standards to the 1 mL extraction solvent to also compensate for losses during extraction.

ANSWER. This point is well taken. We agree that adding the internal standard to the extraction solvent would have been better from a quantitative point of view. We have therefore now modified method section 2.4 to emphasize that data are presenting relative, rather than truly quantitative values. Please, see line 123.

Reviewer #1. 2.4 LC-MS/MS/ analysis

In general I find it very difficult to deduct how the quantification was performed. Generally in LC-MS/MS analyses calibrators, internal controls and samples are prepared in the same way, i.e. you pipette a certain amount of sample and add a certain amount of internal standard to everything. Then the calibration curve is obtained (automatically in the LC-MS/MS system) by plotting the concentration on the X-axis and the response factor (peak area of analyte divided by peak area of internal standard).

In this paper internal standard (unknown amount of 0,25 µM) is added to combined extracts (se comment above too) of tape samples whereas the calibrators are prepared with the internal standard already added. The paper does not mention if calibrators are prepared and aliquoted to be used for different sample sets or if the calibrators are prepared every time samples are run. Or maybe the whole experiment was one single run ? Anyway if samples and calibrators were not prepared on the same day with the same solution of internal standard added in the same amount it could result in erroneous results.

ANSWER. Thank you, reviewer, for these comments. Clarification on sample and calibrators preparation in sections 2.3 Metabolite extraction and 2.4 LC-MS/MS analysis were added. Please, see lines 109-111, 113-117, 141-145.

Briefly, IS were added to reach 0.25 uM concentration in the samples and calibration standards for Phe, Trp and Kyn, and 0.75 uM for Tyr. We used 5uM (Phe, Trp, Kyn) and 15 uM Tyr mixture of IS. Exact volume added to the samples and calibration standards was dependent on the total volumes of the samples. Samples and calibration standards were prepared on the same day using same IS aliquot, and the experiment was done in one single LC-MS/MS run in triplicates.

Reviewer #1. Furthermore - as I read the paper - no internal controls are used. I appreciate that internal controls cannot be prepared in the same way as the samples since you cannot have multiple skin collection sites with known amount of the amino acids. But it would be interesting to see if you could extract a known amount of amino acids that were pipetted directly on the tape (in a small amount of liquid) and then extracted in the same way as the patient samples.

ANSWER. Thanks for this comment. Recovery and matrix effects were examined in a previous study which we now refer to in the method section (lines 109-110). It was noted in the manuscript, that different skin samples, non-lesional skin and melanoma-suspected lesions, were assumed to provide similar and high recoveries. While this can not be assessed in this study, we will refer to the following analysis as relative rather than absolutely quantitative (see lines 119-121):

Reviewer #1. In Table S2 - please adjust the number of digits after the comma to more reasonable numbers.

ANSWER. Thank you for the comment. In S2 Table, the values are rounded to show more reasonable numbers.

Reviewer #1. The precision of the method seems to be a measure of repeatability rather than intermediary precision since it is a measurement of the same calibration solutions three times, as I read it. The precisions obtained are therefore very low and not comparable with intermediary precisions found in the normal way where you have internal controls that are measured together with the calibrators and samples over at least 5 different days from five different sample preparation sets. The limit of quantification would normally be determined as a precision profile showing the lowest concentration that can be measured on internal controls near the LOQ with an intermediary precision below 20% in at least 5 sample sets. Here you do not have any internal controls - hence the calculation from the standard curve instead which may be acceptable.

ANSWER. Thank you for the reviewer’s comment. We agree that in S2 Table, we estimated repeatability rather than precision, which is now clearly stated in S2 Table and main manuscript 2.4. LC-MS/MS analysis section (please, see line 144-145).

Reviewer #1. As regards the table showing the LOD and the LOQ - please explain the two numbers before and after the slashes ?

ANSWER. Thank you for the reviewer’s observations. The LOD and LOQ values following the slashes represent results from a different method, which were mistakenly left in the table after that part of the study was removed from the paper. The values have now been removed.

Reviewer #1. Having said all this the exact concentration may not be very important since extract from the lesions are compared with extracts from the same patient in a healthy area. But you need to describe in more detail, so that we know you are not seeing a systematic difference between lesion type just because the samples were from individual runs with potentially different concentrations of the isotopically labelled internal standards (due to lack of stability). I hope my point makes sense …

ANSWER. Thank you for the comments. As indicated in our response above, we have added more details on the quantitative method used in this study. Only a partial validation of the method was conducted as the aim of the study was to distinguish between non-lesional skin/melanoma lesion rather than to provide highly accurate concentrations for amino acids on skin. Hence, repeatability of the LC-MS/MS analysis was evaluated by running calibration curve standards and samples containing spiked IS in triplicates, as well as all samples and standards were prepared the same day and measured in one LC-MS/MS run.

Reviewer #1. Results

In Table 1 there is a column called Sampling size which shows the diameter defined by adhesive frame. I am not sure how to understand this since earlier in the paper it is stated that the tape is 20 mm in diameter. So why is the sampling size less than 20 mm ?

ANSWER. Thank you, reviewer, for this question. As stated in the 2.1 Materials section diameter of commercially available tapes is 22 mm. However, this does not correspond to the size of melanoma suspected lesions, which varied between the patients, and usually were smaller than 22 mm in diameter. We used a specific sampling frame that matched the size of the melanoma-suspected lesions, as much as possible. A custom-made adhesive frame was employed to define sampling area and minimize the inclusion of non-lesional skin around the melanoma-suspected lesion. This is now clarified in 2.2 Experimental design section (please, see lines 92-94) and Fig. 1. caption (please, see lines 100-103).

Reviewer #1. Figures

Since the LC-MS/MS analysis has not been published previously I think you should add a Figure showing a chromatogram for for instance a calibrator and a patient sample with low concentrations of the analytes.

ANSWER. Thank you for this suggestion. A new figure, showing example of LC-MS/MS chromatograms obtained from standard solution and patient sample was included in Fig 3. Please, see lines 204-205.

Reviewer #2: The study is designed and executed in methodologically and medically sound manner. Clustering of malignant and benign tissues in multivariate analysis is impressive. Discussion on biological reasons of different amino acid signatures is also sound. In conclusion I would recommend this pilot study for publication.

---

## [Decision Letter · Decision Letter 1]

Dear Dr. Jankovskaja,

Thank you for submitting your manuscript to PLOS ONE. After careful consideration, we feel that it has merit but does not fully meet PLOS ONE’s publication criteria as it currently stands. Therefore, we invite you to submit a revised version of the manuscript that addresses the points raised during the review process.

We look forward to receiving your revised manuscript.

Kind regards,

Krishnendu Sinha, Ph.D.

Academic Editor

PLOS ONE

Reviewers' comments:

Reviewer's Responses to Questions

**Comments to the Author**

Reviewer #1: All comments have been addressed

Reviewer #3: All comments have been addressed

2. Is the manuscript technically sound, and do the data support the conclusions?

Reviewer #1: Yes

Reviewer #3: Partly

3. Has the statistical analysis been performed appropriately and rigorously?

Reviewer #1: I Don't Know

Reviewer #3: Yes

4. Have the authors made all data underlying the findings in their manuscript fully available?

Reviewer #1: Yes

Reviewer #3: Yes

5. Is the manuscript presented in an intelligible fashion and written in standard English?

Reviewer #1: Yes

Reviewer #3: Yes

Reviewer #1: The authors have answered all my questions in a satisfying way and I my opinion the paper is ready for publication.

Reviewer #3: The manuscript presents an interesting follow-up study investigating the metabolic alterations of tryptophan (Trp) and kynurenine (Kyn) in melanoma lesions using non-invasive tape sampling. The study builds upon the authors’ previous work and explores whether a higher Kyn abundance and lower Trp abundance can be detected on the surface of cutaneous melanoma lesions compared with adjacent non-lesional skin. While the study is methodologically novel and clinically relevant, several concerns need to be addressed:

1. The authors should clearly define the inclusion and exclusion criteria in the Methods section. Individuals with melanoma or similar skin lesions often have other coexisting dermatological conditions due to shared risk factors such as UV exposure, genetic predisposition, and immunosuppression. These conditions, including actinic keratosis, seborrheic keratosis, dysplastic nevi, and inflammatory skin diseases such as psoriasis or atopic dermatitis, may confound the metabolic profiles observed in the study. The authors should clarify whether such conditions were excluded or controlled for in the study design.

2. It is essential to specify the international classification system used to stratify samples into different groups. Did the study follow the AJCC (American Joint Committee on Cancer) melanoma staging system or the WHO classification of skin tumors? Providing this information would enhance the reproducibility and clinical relevance of the findings.

3. Did the authors observe significant differences in Kyn and Trp levels between:

Malignant melanoma vs. benign pigmented lesions (e.g., dysplastic nevi, seborrheic keratosis)?

Benign lesions vs. adjacent non-lesional skin?

A stratified analysis addressing these comparisons would provide valuable insights into whether Kyn/Trp alterations are melanoma-specific or part of a broader metabolic dysregulation associated with skin lesions.

4. How does the tape sampling technique compare to traditional methods such as punch biopsies and fine-needle aspiration? The authors should discuss the advantages and limitations of tape sampling in terms of sensitivity, specificity, and reproducibility for detecting Trp-Kyn pathway alterations.

5. Given that skin microbiota and pH can influence tryptophan metabolism, it is important to discuss whether microbial degradation of Trp into Kyn (e.g., by skin-resident bacteria such as Cutibacterium acnes) or variations in skin pH might affect the measured ratios. Were any steps taken to mitigate these confounding effects?

6.Higher kynurenine levels have also been reported in psoriasis, an inflammatory skin disease characterized by increased IDO1 expression and Trp metabolism. The discussion should include a comparison of Kyn levels in melanoma versus inflammatory dermatoses such as psoriasis, atopic dermatitis, and lichen planus. This will help differentiate whether increased Kyn is specific to melanoma or part of a broader immunological response in skin disorders.

7. Do kynurenine levels positively correlate with melanoma progression, tumor thickness , ulceration, or metastasis? If such correlations exist, kynurenine could serve as a potential non-invasive biomarker for disease staging and prognosis. The authors should provide statistical analyses supporting or refuting such associations.

8. Several pigmented skin lesions clinically mimic melanoma, including:

• Pigmented Basal Cell Carcinoma (BCC)

• Pigmented Squamous Cell Carcinoma (SCC)

• Atypical/Dysplastic Nevi

• Seborrheic Keratosis

• Blue Nevus

• Lentigo Maligna (a melanoma in situ subtype)

It would be valuable to discuss whether these lesions also exhibit increased kynurenine levels. If available, the authors should compare their findings with existing literature on Trp-Kyn metabolism in these lesions.

9.The study would benefit from a broader contextualization within existing literature. Prior studies using metabolomics approaches have examined Trp-Kyn alterations in melanoma. The authors should integrate findings from studies indexed in PubMed that have investigated Kynurenine as a biomarker for melanoma progression, The role of IDO1/TDO2 inhibitors in melanoma therapy, The impact of immunotherapy (e.g., checkpoint inhibitors) on Trp-Kyn metabolism. A comprehensive discussion comparing the current findings with previous reports would strengthen the manuscript's scientific impact.

10. Given the role of kynurenine in immune suppression via the Aryl Hydrocarbon Receptor (AhR) pathway, could non-invasive kynurenine measurements help in monitoring response to immunotherapy? If kynurenine is a driver of immune evasion, IDO1/TDO2 inhibitors could be potential therapeutic targets. The authors should explore the clinical implications of their findings in melanoma treatment.

Overall, the study provides valuable insights into non-invasive metabolic profiling of melanoma. However, addressing the above concerns will improve the rigor, clinical applicability, and scientific impact of the findings. Providing additional data on stratified comparisons, controlling for confounding factors, and expanding the discussion on kynurenine’s role in melanoma progression and immune evasion would significantly enhance the manuscript.

**Do you want your identity to be public for this peer review?** For information about this choice, including consent withdrawal, please see our Privacy Policy

Reviewer #1: No

Reviewer #3: No

---

## [Author Response · Author response to Decision Letter 2]

10 May 2025

Response to Reviewers have been attached as a separate document.

---

## [Decision Letter · Decision Letter 2]

Non-invasive tape sampling of tryptophan and kynurenine in relation to phenylalanine and tyrosine from melanoma and adjacent non-lesional skin: a pilot study

PONE-D-24-54523R2

Dear Dr. Jankovskaja,

We’re pleased to inform you that your manuscript has been judged scientifically suitable for publication and will be formally accepted for publication once it meets all outstanding technical requirements.

Kind regards,

Krishnendu Sinha, Ph.D.

Academic Editor

PLOS ONE

Additional Editor Comments (optional):

Reviewers' comments:

Reviewer's Responses to Questions

**Comments to the Author**

Reviewer #3: All comments have been addressed

2. Is the manuscript technically sound, and do the data support the conclusions?

Reviewer #3: Yes

3. Has the statistical analysis been performed appropriately and rigorously?

Reviewer #3: Yes

4. Have the authors made all data underlying the findings in their manuscript fully available?

Reviewer #3: Yes

5. Is the manuscript presented in an intelligible fashion and written in standard English?

Reviewer #3: Yes

Reviewer #3: The authors have addressed all the queries satisfactorily and made necessary changes in the manuscript. The manuscript in its present form can be considered for publication.

**Do you want your identity to be public for this peer review?** For information about this choice, including consent withdrawal, please see our Privacy Policy

Reviewer #3: No

---

## [Editor Report · Acceptance letter]

PONE-D-24-54523R2

PLOS ONE

Dear Dr. Jankovskaja,

I'm pleased to inform you that your manuscript has been deemed suitable for publication in PLOS ONE. Congratulations! Your manuscript is now being handed over to our production team.

Kind regards,

on behalf of

Dr. Krishnendu Sinha

Academic Editor

PLOS ONE